# Genetic variations and recurrence in stage III Korean colorectal cancer: Insights from tumor-only mutation analysis

Hajin Jeon[1☉], Jong Lyul Lee[2☉], Hyeran Shim[3☉], Soobok Joe[1], Iksu Byeon[1], Chan Wook Kim[2], Seok-Byung Lim[2], In Ja Park[2], Yong Sik Yoon[2], Hoang Bao Khanh Chu[3], Young-Joon Kim[3]*, Chang Sik Yu[2]*, Jin Ok Yang[1]*

1 Korea Bioinformation Center (KOBIC), Korea Research Institute of Bioscience & Biotechnology (KRIBB), Daejeon, Republic of Korea, 2 Department of Surgery, Division of Colon and Rectal Surgery, University of Ulsan College of Medicine and Asan Medical Center, Seoul, Republic of Korea, 3 Department of Biochemistry, College of Life Science and Biotechnology, Yonsei University, Seoul, Republic of Korea

☉ These authors contributed equally to this work.

* yjkim@yonsei.ac.kr (YJK); csyu@amc.seoul.kr (CSY); joy@kribb.re.kr (JOY)

## Abstract

Colorectal cancer (CRC) has the second highest incidence rate among all cancers in Korea, with approximately 30% of patients with regional CRC experiencing recurrence. Understanding the genetic drivers of recurrence is essential for early detection and targeted treatment. Therefore, many studies have focused on genetic analysis using tumor-normal matched samples, as this approach provides more comprehensive insights. However, tumor-only samples are far more common in clinical practice because of the difficulty in obtaining normal tissues, making developing robust methods for analyzing tumor-only data a pressing need. This study aimed to investigate the genetic variations associated with CRC recurrence using tumor-only whole-exome sequencing data from 200 Korean patients with stage III CRC. By applying stringent filtering using public databases including Genome Aggregation Database (gnomAD), Exome Aggregation Consortium (ExAC), Single Nucleotide Polymorphism Database (dbSNP), 1000 Genomes Project (1000G), Korean Variant Archive 2 (KOVA2), and Korean Reference Genome Database (KRGDB), we identified 221 statistically significant mutations across 195 genes with distinct distributions between the recurrence and non-recurrence groups. Furthermore, statistical analysis of the clinical data revealed that the T-category, N-category, and preoperative carcinoembryonic antigen levels were correlated with CRC recurrence. Moreover, we identified nine networks through protein-protein interaction analysis and identified networks with high feature importance. We also developed a CRC recurrence prediction model using PyCaret, which achieved an area under the curve (AUC) of 0.77. Our findings highlight the importance of robust variant filtering in tumor-only sample analyses and provide insights into the genetic landscape of CRC recurrence in the Korean population.

**Data availability statement:** Data cannot be shared publicly because of legal and ethical restrictions related to the national research program under which the data were generated. The data are scheduled for public release through the Korea BioData Station (https://kbds.re.kr/) under accession numbers KAP220472, KAP220473, and KAP230611. Until public release, access may be requested via the K-BDS platform.

**Funding:** This research was supported by the following grants or programs: "Genomic and other omics data production and analysis" [RS-2024-00438566] from the Ministry of Health and Welfare (MOHW), "Systemic Industrial Infrastructure Projects" [P0009796, 2019] from the Ministry of Trade, Industry, and Energy (MOTIE), the KRIBB Initiative of the Korea Research Council of Fundamental Science and Technology and the Bio & Medical Technology Development Program of the National Research Foundation (NRF) funded by the Ministry of Science & ICT (grant numbers NRF-2017M3A9A7050614 and 2017M3A9A7050610, and 2022R1F1A1074317). The funders had no role in study design, data collection and analysis, decision to publish, or preparation of the manuscript.

**Competing interests:** The authors have declared that no competing interests exist.

## Introduction

Colorectal cancer (CRC) is a significant global health concern, accounting for 10% of the total cancer incidence and 9.4% of cancer deaths [1]. In Korea, CRC poses a major public health challenge and will be the second most commonly diagnosed cancer in both sexes as of 2021 [2]. The annual incidence highlights the criticality of this disease within the national cancer landscape, with CRC representing 12.4% of all new cancer cases and affecting 29,560 individuals [3]. CRC is classified into stages I, II, III, and IV based on the American Joint Committee on Cancer manual [4]. Stage III CRC patients have approximately 1.9 times and 7.8 times recurrence rates, compared to those of stage I and II CRC patients, despite curative tumor resection [5–7]. Although CRC recurrence severely affects patient health and survival, stage III CRC patients with relatively high recurrence rates undergo a similar process of adjuvant chemotherapy and surveillance, regardless of genetic risk. This high recurrence rate underscores the importance of understanding the genetic differences between recurrent and non-recurrent CRC. Current research efforts have focused on investigating the recurrence risks associated with stage III CRC, highlighting a wide range of analyses to better understand and manage this disease [8,9].

Several researchers use the gold standard method of identifying cancer-specific somatic mutations by comparing tumor tissues with normal tissues. However, using normal samples can be impractical due to logistical, financial, and sample availability constraints. In such scenarios, the reliance on tumor-only mutation detection is essential. Indeed, the College of American Pathologists reported that 95% of laboratories performed NGS-based testing of tumor-only samples [10]. Unlike the typical gene-level determination of the presence of mutations, tumor-only analysis may show poor precision owing to the detection of germline variants. To address this issue, it is essential to filter out common variants from various large-population databases [11] and analyze them at the mutation level [12]. This approach reduced the likelihood of coincidental overlapping mutations across multiple samples. In fact, the recall rate of the tumor-only analysis is very similar to that of the tumor-normal analysis [12].

While large population databases can be used to filter out common germline variants, tumor-only sequencing inherently cannot distinguish somatic mutations from rare germline variants. It is therefore possible that germline mutations remain in the dataset. One commonly used method to estimate this is by analyzing the variant allele frequency (VAF), as heterozygous germline variants are typically observed with VAFs between 40% and 60% in diploid regions [11,13].

Protein-protein interactions are crucial to every cellular process, including cell cycle control, splicing, and regulation of gene expression levels [14,15]. These interactions provide insights into the functional dynamics of genes and proteins and aid in predicting potential cancer-associated genes [16]. Ingenuity Pathway Analysis (IPA) is a widely used software for understanding proteomic data and is capable of constructing protein-protein interaction networks. This tool integrates comprehensive data from a curated knowledge base, research publications, and various third-party

databases to derive meaningful interpretations [17,18]. Networks obtained from IPA are actively used to study the progression, metastasis, and recurrence of CRC [19–23]. Additionally, various studies have incorporated machine learning algorithms, such as support vector machine and random forest model, into these networks to build predictive models [24,25].

In this study, we focused on Korean patients diagnosed with stage III colorectal cancer, as this subgroup accounts for a substantial portion of CRC-related mortality in Korea. Given the relatively homogeneous genetic background of the Korean population and the high recurrence burden observed in stage III cases, this focus allows for a clearer investigation of recurrence-associated characteristics. We proposed and validated the use of multiple public population databases to filter out inherited variants in tumor-only samples. We also determined the relationships among the genes containing these mutations through protein-protein interaction networks and implemented a CRC recurrence prediction model using these networks.

## Materials and methods

### Ethics statement

The study protocol was approved by the Institutional Review Board of Asan Medical Center (registration number: 2017–1191) and was conducted per the Declaration of Helsinki. Normal tissues used for variant validation were collected from patients receiving treatment at Asan Medical Center in Korea as part of the Integrated Multi-Genomics-based Precision Medicine in Colon Cancer Project. This research adhered to ethical standards, and tissue sample usage was approved by the Yonsei University Institutional Review Board (IRB No. 7001988–202010-BR-727–02). All participants provided written, informed consent at the time of sample collection. The raw data were deposited in the Korean Nucleotide Archive (KoNA) [26] under accession IDs KAP220472, KAP220473 and KAP230611. All samples and medical data used in this study have been irreversibly anonymized. The samples were screened from October 20, 2017 to October 19, 2018, and the data were accessed for research purposes from April 1, 2024, to August 2, 2024.

### Sample preparation

This study retrospectively included 200 patients with CRC from Asan Medical Center, Seoul, Korea, between 2011 and 2016. The CRC samples included 200 fresh frozen CRC tissues and were provided by Asan Bio-Resource Center, Korea Biobank Network (2017-14(152)). The study cohorts were enrolled when the CRC was pathologically confirmed to be stage III adenocarcinoma after curative-intent surgery and the tumor tissue could be resected to secure sufficient tissue samples for this study. Among the study cohorts, patients who underwent preoperative chemoradiotherapy or neoadjuvant chemotherapy and those with hereditary CRC or other malignancies within 5 years of surgery were excluded. All enrolled patients underwent regular follow-ups with laboratory and imaging examinations. During regular follow-ups, when recurrence was suspected, recurrence was defined when histologic confirmation was made by biopsy of the suspected lesion. If histological confirmation was not available, sequential enlargement of the suspected recurrent lesion observed in radiological studies or positive findings on positron emission tomography were used to confirm recurrence. As this was a retrospective study, it was conducted by randomly selecting 100 patients each from the recurrence and non-recurrence groups, while the recurrence group could be identified.

Normal colorectal and tumor tissues were minced into small pieces and frozen in sterile culture dishes. Genomic DNA was extracted from tissue fragments using the DNeasy Blood & Tissue Kit (Qiagen), with slight modifications to the manufacturer's protocol. The quantity and quality of the purified genomic DNA were assessed using the Quant-iT PicoGreen dsDNA assay (Invitrogen) and TapeStation gDNA ScreenTape (Agilent), respectively.

### Sequencing and data processing

Whole-exome sequencing was performed using 50 ng of input genomic DNA, with target enrichment and library preparation using the Twist Library Preparation EF Kit (Twist Bioscience, PN 101058) and full-length combinatorial dual-index TruSeq-compatible Y-adapters (Illumina). Libraries were prepared according to the Twist Bioscience Library Protocol.

The final purified sequencing libraries were quantified by qPCR, following the KAPA Library Quantification Protocol Guide (KAPA Biosystems) for Illumina platforms, and qualified using TapeStation DNA ScreenTape D1000 (Agilent). Sequencing was performed using a NovaSeq platform (Illumina).

For whole-genome sequencing, sequencing libraries were prepared using the TruSeq DNA Nano Library Prep Kit (Illumina) per the manufacturer's guidelines. Purified libraries were quantified by qPCR according to the KAPA Library Quantification Protocol Guide (KAPA Biosystems) for Illumina sequencing platforms and assessed using TapeStation (Agilent). Sequencing was performed using the NovaSeq platform (Illumina).

We processed sequence data obtained from whole-exome and whole-genome sequencing by aligning them to the human reference genome GRCh38 using Burrows-Wheeler Aligner (BWA) version 0.7.17 [27]. Post-alignment realignment and quality recalibration were performed using the Genome Analysis Toolkit (GATK) [28]. Variants at each chromosomal position were identified using HaplotypeCaller in GATK. The mutations were annotated using ANNOVAR [29] for human genome build 38. To filter out common variants, we compared our data against East Asian populations using multiple databases: gnomAD version 4.0 (both whole-exome and whole-genome data) [30], ExAC 65000 exome allele frequency data (ExAC) [31], dbSNP150 with allelic splitting and left-normalization (dbSNP) [32], and 1000 Genomes data based on the 201508 collection v5b (1000G) [33].

## Variant and sample filtering

In this study, we focused our analysis on coding regions using whole-exome sequencing data. Non-coding areas were filtered out, and variants located in the splicing sites and exonic regions were selected. This approach retains genetic variations including splicing variants, frameshift indels, in-frame indels, nonsense variants, and non-synonymous SNVs.

To ensure data quality and relevance, we filtered out the variants identified as N/A in more than 30% of the samples. Common genetic variations with allele frequencies of > 5% were eliminated. We used ANNOVAR to obtain allele frequencies in the East Asian population, as indicated by the gnomAD, ExAC, and 1000G databases. For the localized approach, we used the Korean Variant Archive 2 (KOVA 2) [34], filtering out variants with an AF of over 5% within the Korean population. By removing common genetic variants using existing databases, we aimed to detect somatic tumor mutations without matching normal samples.

When removing common variants in the Korean population, we used KOVA and the Korean Reference Genome Database (KRGDB) [35]. To obtain the KRGDB allele frequency, we matched each variant to dbSNP to obtain rsIDs and then accessed the KRGDB allele frequency information stored in dbSNP because direct access to KRGDB is currently unavailable. Furthermore, we found that an official API through Entrez Programming Utilities did not provide sufficient information, as provided on the dbSNP webpage, and we developed a script available at https://github.com/Hajin-Jeon/KRGDB-parser. We used this script only after finding statistically significant mutations to minimize the load on the NCBI server because it accesses webpages directly.

We analyzed the mutation profile of the filtered dataset to identify samples with an abnormal number of mutations, indicative of hypermutation. A higher-than-usual tumor mutational burden (TMB) cutoff of 24 variants per megabase was applied, acknowledging the increased likelihood of false positives in tumor-only samples. Samples exceeding this threshold were excluded from the primary analysis but were subjected to a separate analysis to explore their mutation characteristics.

After filtering variants and samples, we determined the frequency of amino acid substitutions caused by non-synonymous SNVs. This analysis enabled us to create a map of these substitutions, helping to identify changes that are harmful to CRC.

## Statistical analysis of clinical data

After sample filtering, the following clinical variables were analyzed: sex, age, preoperative carcinoembryonic antigen (CEA) level, tumor location, operation name, tumor size, Bormann type, harvested lymph node number, T-category, N-category, pathological stage, differentiation, lymphovascular and perineural invasion, adjuvant chemotherapy, and

adjuvant radiotherapy. We conducted an Analysis of Variance (ANOVA) to evaluate the impact of the selected variables in the clinical data on the likelihood of CRC recurrence. At this time, a *P*-value less than 0.05 was considered statistically significant, indicating a strong association with recurrence outcomes. For variables that showed a significant relationship with recurrence in the ANOVA, we performed Ordinary Least Squares (OLS) regression to determine the precise quantification of the influence of each significant factor.

## Statistically significant mutation analysis

In this study, we defined a Statistically Significant Mutation (SSM) based on the odds ratios (ORs) calculated for each genetic variant to identify mutations with significant frequency differences between the recurrence and non-recurrence groups. To reduce the bias, we adjusted each cell by adding 0.5 [36]. We used the 'orscoreci' function from the R package PropCIs and excluded any variants whose 95% confidence intervals included the value of 1 [37].

We investigated the biological significance of the remaining variants by determining their presence within known protein domains using the UniProtKB database [38]. Variants that were not associated with any documented protein domains were excluded. Subsequently, we explained the molecular interactions by generating a protein-protein interaction network for genes containing SSMs using Ingenuity Pathway Analysis (IPA) [18]. These interactions were visualized using Cytoscape [39], providing insights into the potential biological impacts of our findings. To investigate the effects of SSMs at splice sites on transcription, we utilized SpliceAI [40,41] and SpliceAPP [42] to assess their potential impacts. The nucleotide and protein sequences of the exons affected by these splice site mutations were extracted from the NCBI database. Additionally, protein sequences were modeled using AlphaFold2 [43] to compare structures that included or excluded the affected exons. Integrative Genomics Viewer (IGV) [44] and Mol* [45] were used to visualize mutation information and protein structures.

We developed our prediction model based on the networks identified using IPA. To construct the model, we utilized Python version 3.8.17 and PyCaret version 2.3.10. Each sample was represented by features corresponding to the number of genes within each identified network, including those associated with the SSMs. We employed various models available in PyCaret and selected the model with the highest area under the ROC curve (AUC). Our model training utilized a 10-fold cross-validation technique with inputs normalized using the min-max algorithm. The selected model was then evaluated for feature importance and ROC curve effectiveness to provide a comprehensive understanding of its predictive power.

## Variant validation

In this study, we excluded common genetic variants from tumor-only samples using various databases, including gnomAD, ExAC, 1000G, KOVA, and KRGDB. Despite our thorough filtering steps, these resources cannot fully guarantee the elimination of all germline variants, indicating that false-positive variants can be included. To validate the effectiveness of the filtering process, we utilized normal samples from a cohort of Korean patients with stage III CRC from the same hospital, who were expected to have a similar set of common variants. We analyzed 68 whole-genome sequencing samples using the same pipeline as that for the whole-exome sequencing data. To save time, we did not analyze all whole-genome sequence regions but only the genomic regions covered by whole-exome sequencing data.

Here, we evaluate the effectiveness of third-party databases (gnomAD, ExAC, 1000G, KOVA, and KRGDB) compared to normal samples from the comparable cohort based on the following criteria: A variant that was not classified as a common variant by both the databases and normal samples is assumed to be a germline variant; a variant classified as a common variant by both the databases and normal samples is assumed to be a somatic variant; a variant classified as a common variant by the databases, not by normal samples is considered as a false positive in our methodology, which means minimizing these instances is crucial; a variant was not classified as a common variant by the databases but classified as a common variant by normal samples suggests that using databases is more effective for variant filtering than

relying solely on similar cohort data, though it also indicates a higher risk of false negative. To evaluate the effectiveness of third-party databases for variation filtering, we counted the number of false-positive variants; a lower number indicated higher effectiveness. Additionally, we calculated the precision and recall by comparing the variants from the databases with variants from the normal samples.

We also assessed whether the identified somatic single nucleotide mutations (SSMs) might include germline variants by calculating the variant allele frequency (VAF) for each mutation. For each SSM, we determined the proportion of samples with non-zero VAFs that fell within the 40–60% range, which corresponds to the expected range for heterozygous germline variants in diploid regions. This allowed us to estimate the likelihood that each SSM represents a germline mutation.

## Results

In this section, we present a comparative analysis between the recurrence and non-recurrence groups, following the methodologies outlined in the Methods section, using data from 200 patients with pathological stage III CRC. Owing to missing read pairs, 14 samples were excluded, resulting in 186 samples for further analysis. Initially, non-target variants and outlier samples were excluded using ANNOVAR in conjunction with existing databases. We constructed a mutation profile based on the variant frequencies using the curated dataset. This profile provides an overview of the distribution and prevalence of various mutations in recurrence and non-recurrence groups. We filtered out the hypermutated samples for subsequent analyses, leaving 173 samples. Following frequency-based mutation profiling, we focused on mutations in known protein domain regions. The identified mutations were further analyzed for protein-protein interactions using IPA and visualized using Cytoscape. This study aimed to elucidate the functional dynamics of these mutations and their impact on CRC recurrence.

### Variant and sample filtering

Before sample filtering, we analyzed tumor-only samples from 186 patients with CRC, comprising 91 patients in the recurrence group and 95 in the non-recurrence group. After aligning the sequence data, we achieved an average sequencing depth of 76.17X (range, 67-87X) for the target regions (S1 Table). Using ANNOVAR, we annotated and identified 4,911,291 variants. After filtering 305,331 non-coding region variants, 4,605,960 variants remained in the coding regions. As shown in Fig 1, among the variants in the coding regions, we excluded 2,176,669 variants from the intergenic regions, 2,030,169 from the intronic regions, 110,736 from the UTRs, and 84,275 from the unclassified regions. This filtering

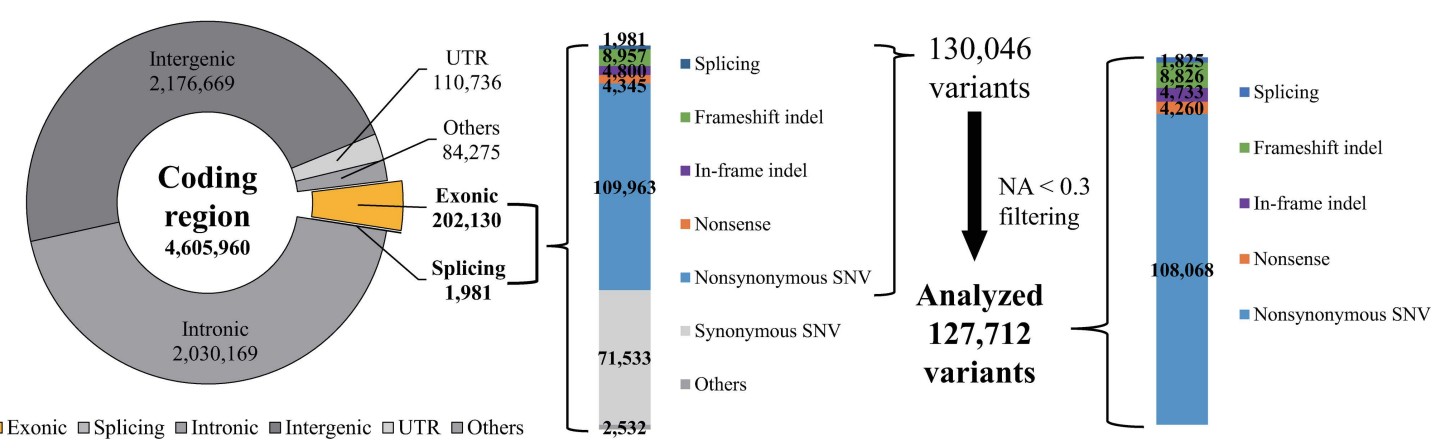

**Fig 1. Variant filtering in the coding region.**

process left 202,130 variants in the exonic regions and 1,981 in the splicing sites. After selecting only specific types (splicing sites, frameshift indels, in-frame indels, nonsense variants, and non-synonymous SNVs), 130,046 variants remained. Filtering out variants occurring as N/A in over 30% of the samples resulted in 127,712 variants.

We filtered somatic mutations using existing databases, including gnomAD, ExAC, 1000G, and KOVA to highlight somatic mutations. Using these databases, 34, 260 common variants were excluded. Fig 2 shows the number of common variants in each database. As shown in Fig 2A, gnomAD uniquely identified 13,671 cases, highlighting its extensive coverage. Fig 2B compares two gnomAD datasets: whole-exome and whole-genome. It was found that 58.5% (19,232 of 32,891) of the variants were present in both datasets. However, 39.2% (12,883 of 32,891) were exclusive to whole-exome data and 2.4% (782 of 32,891) were exclusive to whole-genome data.

Particularly, when comparing gnomAD versions 2.1.1 and 4.0.0, filtering with version 2.1.1 resulted in 25,361 mutations being filtered out. Differences between versions affected the discovery of common variants. For example, the in-frame insertion variant LNP1:c.194_195insTCCTAGAAGGCATTCTCATGAGGACCAGGAATTCCGATGCCGATCGTCTGAC-CGTCT: p. S80_H81insSDRLPRRHSHEDQEFRCRS, which appeared in 90.8% of the analyzed samples. Because this variant was not present in version 2.1.1, this result suggests that it could be a specific Korean CRC variant. However, using version 4.0.0, it was found to have an allele frequency of 0.7143 in the East Asian population, which is one of the common variants; therefore, it was filtered out.

Besides variant filtering, we counted the number of mutations in each sample to identify and exclude the outliers. Fig 3 shows the mutation frequencies in 186 CRC samples. We applied a higher-than-usual TMB cutoff of 24 mutations per megabase, which clearly separated hypermutated tumors from non-hypermutated tumors. At this stage, we counted only the mutations included in the 93,452 previously mentioned variants. As a result, we excluded 5 samples from the recurrence group and 8 from the non-recurrence group, leaving a total of 173 samples. Subsequent analyses were performed using a curated dataset.

## Statistical analysis of clinical data

Table S2 shows the number of patients for each clinical variable with mean and standard deviation, which contain both values before sample filtering (n = 200) and after sample filtering (n = 173). Table 1 shows the ANOVA results for the selected clinical variables (sex, age, T-category, N-category, pathological stage, tumor location, preoperative CEA level,

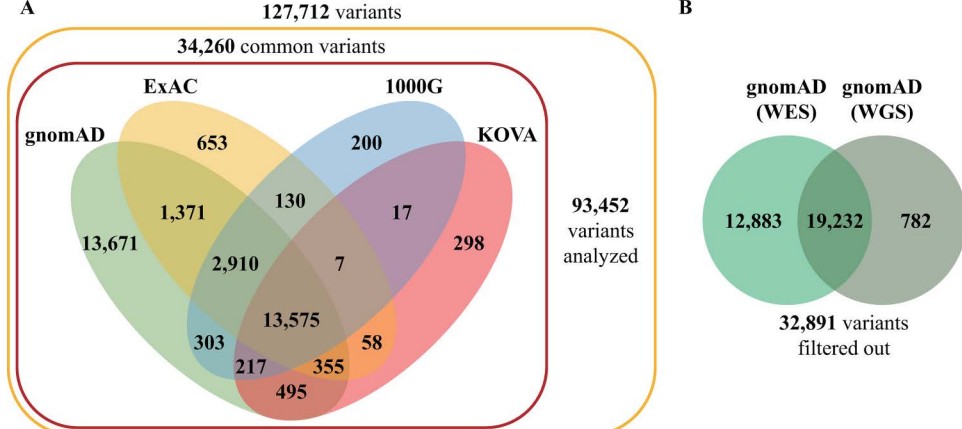

**Fig 2. Number of genetic variants filtered out by each database. (A) The number of common variants in each database. (B) The number of common variants in gnomAD datasets.**

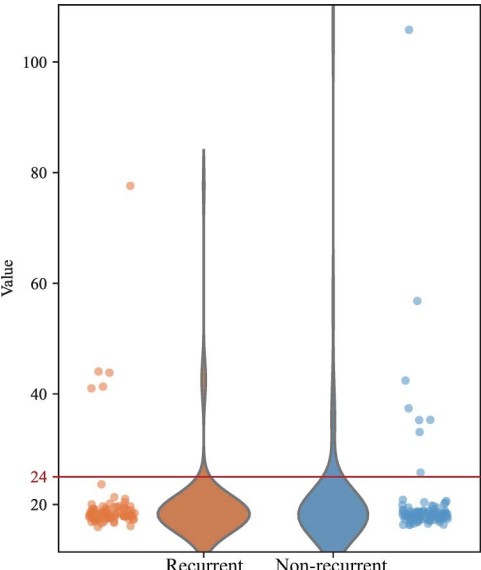

**Fig 3. Variant frequencies in human colorectal cancer.**

and adjuvant chemotherapy regimen). As indicated in the table, the T-category, N-category, and preoperative CEA levels demonstrated a significant relationship with CRC recurrence *P*-value less than 0.05.

In the regression analysis, significant associations were identified between certain clinical variables and CRC recurrence. Table 2 shows the relationships between the variables and recurrence risk using OLS results. The T-category showed a statistically significant positive correlation with recurrence risk, with a coefficient of 0.1937, indicating that a higher T-category was associated with an increased likelihood of recurrence. The N category showed a less positive effect, with a coefficient of 0.0547. The preoperative CEA level demonstrated a coefficient of 0.0054 with a standard error of 0.002. This low standard error, combined with a statistically significant *P*-value of 0.021, highlights the strong and reliable predictive power of CEA levels for CRC recurrence.

Adjusted R-squared, 0.07; F-statistic, 5.332; Prob > F (*P*-value of the entire model), 0.00156.

Fig 4 illustrates the effect of each variable on CRC recurrence. In Figs 4A and 4C, values close to 0.0 indicate belonging to a non-recurrence group, while values close to 1.0 indicate a recurrence group. Error bars represent 95% confidence

**Table 1. Result of ANOVA for the selected clinical variables.**

|  | Sum of squares | F-value | P-value |
|---|---|---|---|
| **Sex** | 0.038759 | 0.153386 | 0.695808 |
| **Age** | 0.647986 | 2.601035 | 0.108638 |
| **T-category** | 1.549159 | 6.352759 | 0.012635 |
| **N-category** | 1.100746 | 4.465893 | 0.036029 |
| **Pathological stage** | 0.861678 | 3.476239 | 0.06397 |
| **Tumor location** | 0.645268 | 1.287406 | 0.278662 |
| **Preoperative CEA level** | 1.045247 | 4.235148 | 0.041113 |
| **Adjuvant chemotherapy regimen** | 1.782175 | 2.430884 | 0.067138 |

**Table 2. Result of OLS for the selected clinical variables.**

| Variable | Coefficient | Standard Error | t-value | P>|t| | Confidence Interval (95%) |
|---|---|---|---|---|---|
| Constant | -0.2902 | 0.245 | -1.183 | 0.239 | [-0.775, 0.194] |
| T-category | 0.1937 | 0.075 | 2.586 | 0.011 | [0.046, 0.342] |
| N-category | 0.0547 | 0.027 | 2.01 | 0.046 | [0.001, 0.108] |
| Preoperative CEA level | 0.0054 | 0.002 | 2.326 | 0.021 | [0.001, 0.01] |

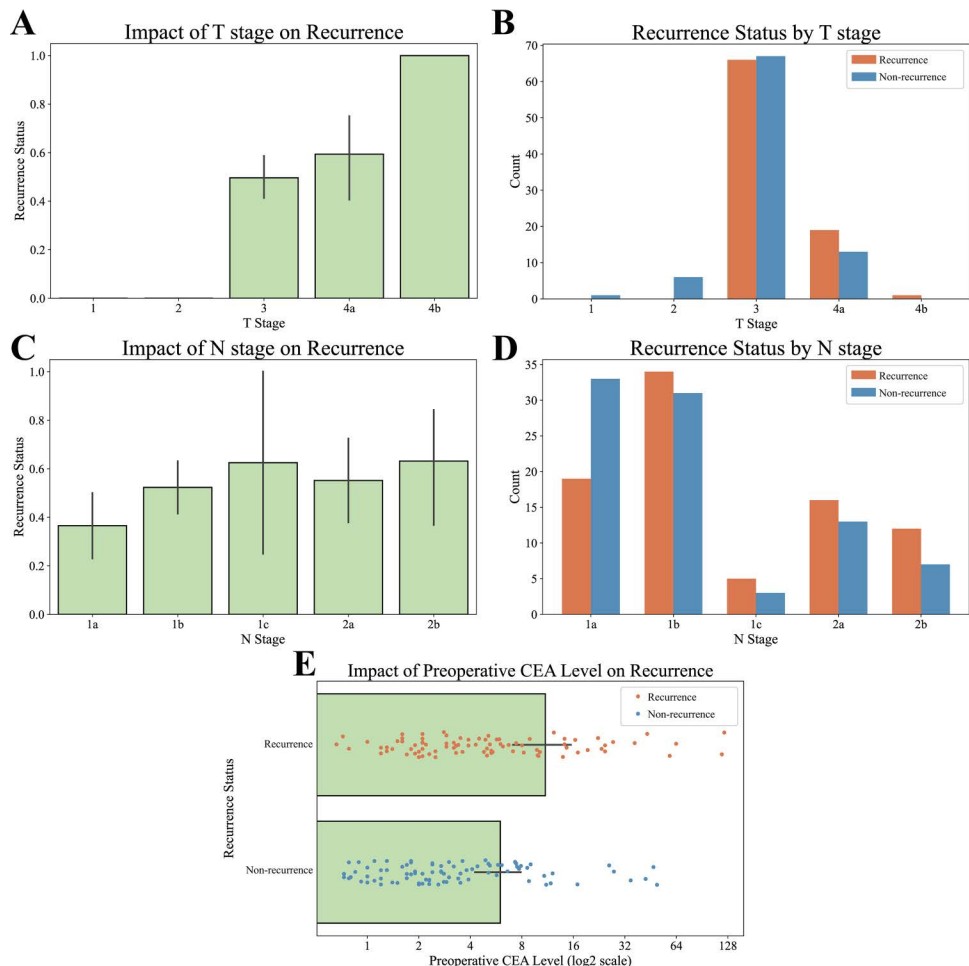

**Fig 4. Impact of clinical variables.**

intervals. As shown in Figs 4A and 4B, most samples were categorized as T3, as the pathological stage for all samples was uniform stage III. All patients in the T1 and T2 groups were classified into the non-recurrence group. However, at T4, more patients were in the recurrence group than in the non-recurrence group. Figs 4C and 4D reveal that the number of individuals in the non-recurrence group at N1a significantly exceeded that in the recurrence group. However, for N1b, N1c, N2a, and N2b, every case in the recurrence group was higher than in the non-recurrence group. Fig 4E shows that higher preoperative CEA levels were associated with an increased likelihood of inclusion in the recurrence group.

## Mutation profiles

We identified 93,452 mutations with an average of 1,051.2 mutations per sample. Notably, the recurrence group displayed an average of 8.1 (0.8%) more mutations than did the non-recurrence group. The recurrence group had a TMB of 17.4, whereas the nonrecurrence group had a TMB of 17.3.

For a detailed analysis, we categorized the mutations into five types: nonsynonymous SNVs, frameshift indels, in-frame indels, nonsense mutations, and splicing sites. Figs 5 and S1 show the mutation spectra of both groups after removing outliers. Table A in S1 File summarizes the mutation profile, whereas Table B in S1 File presents the detailed profiles of each group and mutation type. Figs A and B in S1 show a plot of the mutation spectra of each group. Our analysis revealed no significant differences in base substitutions, insertions, or deletions within the same mutation type between the two groups.

However, distinct patterns were observed in the spectra of mutation types (Fig 5). Among nonsense mutations, C:G>T:A and C:G>A:T were more frequent than the average mutation frequencies, whereas C:G>G:C and A:T>G:C were less frequent. Furthermore, indels in the splicing regions demonstrated a high frequency, with insertions accounting for approximately 90% of these indels, which was a significant increase from the total average of less than 40%.

The bar plot on the right side of Fig 5 shows the average number of mutations per sample. While the differences in non-synonymous SNVs, nonsense mutations, and splicing sites between the recurrence and non-recurrence groups were less than one mutation per individual, the recurrence group showed a higher number of mutations: 3.4 (6.7%) more in frameshift indels and 4.9 (7.5%) more in in-frame indels than in the non-recurrence group. A detailed comparison between the groups is shown in Figs C and D in S1 File.

We analyzed amino acid substitutions using 77,178 nonsynonymous SNVs (the value of missense mutations of the overall group, as shown in Table A in S1 File) among the 93,452 overall mutations. The most common amino acid substitutions were R>Q (3,439, 4.5%), R>H (3,271, 4.2%), A>T (3,056, 4.0%), R>C (2,813, 3.6%), and A>V (2,803, 3.6%) (S1 Fig).

We also analyzed the 13 excluded hypermutated samples with a TMB greater than 24 mutations per megabase: 5 in the recurrence group and 8 in the non-recurrence group. While this subset was too small for robust statistical comparisons, we observed distinct patterns in their mutation spectra compared to the non-hypermutated samples. For hypermutated samples, Table C in S1 File presents the overall mutation profiles, and Table D in S1 File shows the detailed breakdown by group and mutation type. Figures E and F in S1 File display the mutation spectra of individual samples in

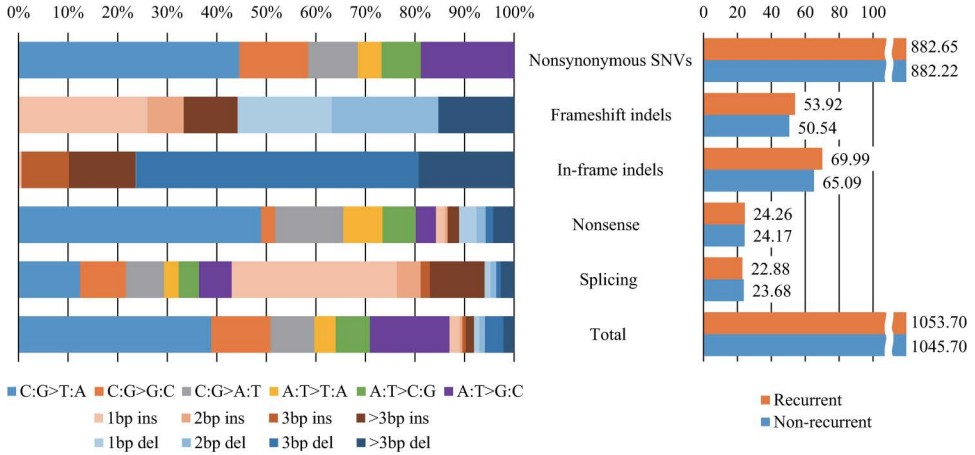

**Fig 5. Mutation spectrum of the data samples.**

each group, sorted by TMB. Figures G and H in S1 File compare the mutation characteristics between recurrence and non-recurrence groups within the hypermutated subset.

## Statistically significant mutations and protein-protein interaction

We identified 441 SSMs by comparing the recurrence and non-recurrence groups using odds ratios, we initially identified 441 SSMs. Upon reviewing each mutation using UniProt, we discovered that 241 (55.5%) were located within the domain regions. Additionally, we checked allele frequencies for these mutations using values from the KRGDB registered in the dbSNP and filtered out mutations with frequencies greater than 0.05. In total, 221 SSMs (S3 Table).

Tables 3 and 4 provide detailed profiles of the mutations. Table 3 shows the exonic functions of the mutations, whereas Table 4 shows the distribution of the mutated genes across the groups. Among the genes with domain region mutations, 101 were predominantly found in the recurrence group, 91 in the non-recurrence group, and three in both groups.

We analyzed seven SSMs in splicing sites in detail through existing splice site analysis tools. Using SpliceAI, we found that two of these mutations significantly contributed to splice loss. Specifically, a mutation in *LILRA2* (chr19–54574075-G > C, rs1461200853 in the dbSNP) was associated with a high probability of splice loss (delta value = 0.92) due to acceptor loss (delta value = 0.86). Similarly, a mutation in *SRPRA* (chr11–126267174-C > G) resulted in donor loss (delta value = 0.94) and a corresponding splice loss (delta value = 0.99). S4 Table displays the mutation information, odds ratios, and SpliceAI model delta scores for these seven SSMs at the splicing sites as well as the nucleotide and protein sequences of the affected exons and introns where splice loss occurs. Using SpliceAPP for SNVs, we found that a mutation in *LILRA2* could affect the 3' splice site, consistent with SpliceAI's prediction. Similarly, a mutation in *SRPRA* was identified at the 5' splice site and was predicted to have a significant impact, with a score of 0.9307, indicating a high likelihood of disrupting canonical splicing. SpliceAPP could not find significant effect at the mutation in *SERPINB6* (chr6–2970757-G > T) as SpliceAI. S5 Table shows the result of SpliceAPP. Additionally, we visualized the effective splicing site mutations with their surrounding regions in all samples using IGV (Figs A and B in the S2 File). The exon skipping and intron retention structures, the most common forms of alternative splicing from splice loss, were predicted using Alpha-Fold2 and visualized using Mol* (Figs C, D, E, F, G, and H in the S2 File).

**Table 3. Number of SSMs for each exonic function.**

|  | Initially identified SSMs | SSMs within domain regions | SSMs after KRGDB filtering |
|---|---|---|---|
| **SNV** | 365 | 199 | 172 |
| **Frameshift indel** | 24 | 14 | 14 |
| **In-frame indel** | 37 | 24 | 24 |
| **Nonsense** | 8 | 4 | 4 |
| **Splicing site** | 7 | N/A[a] | 7 |
| **Total** | 441 | 241 | 221 |

[a]Splicing sites are not classified within domain regions because they are not in the exonic region.

**Table 4. Number of genes that contain SSMs for each group.**

|  | Genes |
|---|---|
| **Recurrence group** | 101 |
| **Non-recurrence group** | 91 |
| **Both groups** | 3 |
| **Total** | 195 |

Subsequently, we analyzed protein-protein interactions using IPA for 195 genes. Nine networks comprising at least two genes. Table 5 lists the functions of each network detected by IPA. The figures and gene lists for each network are presented in File S.

PyCaret was used to evaluate the various models (S6 Table). Because PyCaret does not provide the AUC for Support Vector Machine (SVM) and ridge regression models, we calculated these values using the sci-kit-learn library. After evaluating the models, we selected the logistic regression model because it showed the highest AUC. The model's feature importance and ROC curves after tuning are shown in Fig 6. Fig 6A indicates that networks 8 and 5 have high variable importance, and Fig 6B shows that both networks have the same high performance in predicting the group (AUC = 0.77). Figs 6C and 6D illustrate the networks with the greatest impact according to feature importance, which are the same as Figs E and H in S3 File.

### Variant validation using comparable cohort

Table 6 shows the confusion matrix for the variants classified by third-party databases compared with the normal samples of the comparable cohort. Here, we assume that the normal samples are in an actual condition because they have similar conditions; therefore, a similar variant set is expected. In the table, the sum of the values in the database-true column is the number of variants that we analyzed in this study, and the sum of the values in the database-false column is the number of variants that we classified as common variants. The confusion table showed a precision of 0.973 and a recall of 0.845.

For the aspect of the possibility of germline mutation, we found that 100 out of the 221 SSMs showed a greater than 50% likelihood of being germline mutations through calculating VAF. These results are summarized in S7 Table, which presents the estimated germline mutation probability for each SSM across sample groups.

## Discussion

In this study, we analyzed the tumor-only whole-exome sequencing data from 200 Korean patients with stage III CRC. Owing to the absence of normal tissue data for comparison, we leveraged existing genetic variant databases to identify and exclude common variants. Subsequently, we examined the mutation profiles to identify SSMs and genes, which facilitated our analysis of protein-protein interactions, leading to the identification of nine distinct networks.

Our findings indicate that certain amino acid substitutions are consistent with those previously identified in CRC but not in other cancer types [46]. Specifically, the substitutions R > Q, R > H, A > T, R > C, and A > V identified in our study mirror the top 5 common substitutions reported in the literature (S1 Fig). This consistency underscores the robustness of our sample collection and bioinformatic analysis protocols.

**Table 5. Main functions of each protein-protein interaction network.**

| No. | Main function of the network |
|---|---|
| 1 | Dermatological Diseases and Conditions, Immunological Disease, Organismal Injury and Abnormalities |
| 2 | Cancer, Hematological Disease, Organismal Injury and Abnormalities |
| 3 | Inflammatory Response, Neurological Disease, Organismal Injury and Abnormalities |
| 4 | Cell Cycle, Cellular Assembly and Organization, DNA Replication, Recombination, and Repair |
| 5 | Gastrointestinal Disease, Hepatic System Disease, Organismal Injury and Abnormalities |
| 6 | Cell-to-cell signaling and Interaction, Cellular Assembly and Organization, Renal and Urological System Development and Function |
| 7 | Cell-to-cell signaling and Interaction, Hematological System Development and Function, Immune Cell Trafficking |
| 8 | Amino Acid Metabolism, Cellular Development, Cellular Growth and Proliferation |
| 9 | Cancer, Cell Cycle, Gene Expression |

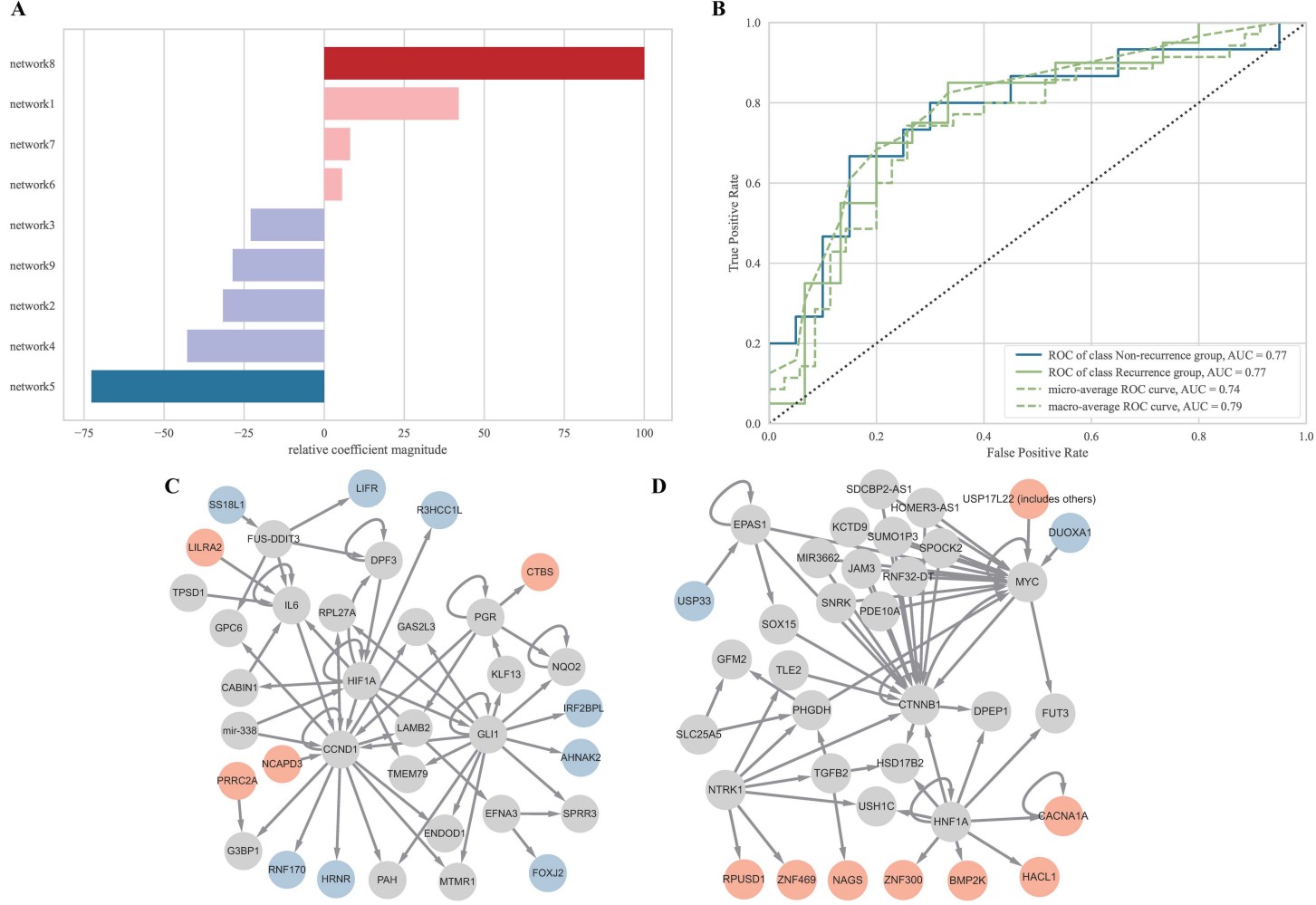

**Fig 6. Model evaluation and feature importance analysis. (A) Feature importance of the model. (B) ROC curves of the model after tuning. (C) Protein-protein interaction network 5. (D) Protein-protein interaction network 8.**

**Table 6. Confusion matrix for the uncommon variants classified by databases (i.e., predicted condition) compared to normal samples (i.e., assumed actual condition).**

|  | Database - True | Database - False |
|---|---|---|
| **Normal - True** | 90,908 | 16,704 |
| **Normal - False** | 2,544 | 17,556 |

Even within the same population, mutation frequencies can vary significantly across different genetic variant databases, owing to differences in datasets, sequencing methods, and quality control criteria. As shown in Fig 2A, our analysis revealed distinct common variants in East Asians across the gnomAD, ExAC, and 1000G databases. This variability highlights the importance of using multiple genetic variant databases to accurately filter inherited variants from tumor-only samples [12]. However, this approach prioritizes reducing false positives such as retained germline variants over maximizing sensitivity and accepts the trade-off that some true somatic mutations may be inadvertently excluded. We deliberately

adopted a conservative filtering strategy to minimize false positives, even at the cost of accepting a higher risk of false negatives. Although public databases may not fully represent the Korean population and may contain inaccuracies, using multiple sources in combination helps mitigate the limitations of any single dataset and supports more reliable interpretation of tumor-only mutation calls.

When comparing the whole-exome and whole-genome databases within gnomAD (Fig 2B), we identified more common variants using the whole-exome database, as expected. However, the inclusion of whole-genome data identified an additional 782 common variants, suggesting that both databases should be utilized even when only whole-exome sequencing data are available. Furthermore, the gnomAD whole-exome database version 4.0 identified 26.6% more common variants than version 2.1.1, emphasizing the need to use the most updated versions of genomic databases [47,48].

For the Korean CRC samples, we used the KOVA database, which is a reference for genetic variation in the Korean population. Despite this, KOVA identified fewer common variants (15,022) than our databases, such as gnomAD, ExAC, and 1000G. Additionally, the KRGDB faced accessibility issues and potential mapping inaccuracies due to its hg19 genome build foundation [49,50]. These findings highlight the need for a more comprehensive and accessible Korean reference database.

After filtering, the mutation profiles of the 173 retained samples were analyzed. As shown in Fig C in S1 File, no significant differences were observed in the mutation spectra between the groups. However, frameshift and in-frame indels were more frequent in the recurrence group than in the non-recurrence group, by 6.7% and 7.5%, respectively (Fig 5 and Fig D in S1 File). Notably, the in-frame mutation in *MUC4* was more prevalent in the non-recurrence group (OR = 0.07), which aligns with research linking this mutation to a better overall survival prognosis [51]. Conversely, a frameshift mutation in *HLA-A* was more frequent in the recurrence group (OR = 11.8), supporting studies suggesting that *HLA-A* function loss aids cancer progression through immune evasion [52,53].

Although hypermutated tumors were excluded from the main analysis due to their small number, we recognize their potential biological relevance. To address this, we analyzed the mutation profiles of the 13 hypermutated cases, which exhibited distinct patterns compared to the non-hypermutated cohort. The results of this exploratory analysis are provided in Tables C and D in S1 File and Figures E to H in S1 File. However, due to the limited sample size, these differences are not statistically conclusive. Further investigation using larger hypermutated cohorts will be necessary to clarify the role of hypermutation in CRC recurrence.

Through the analysis of clinical data among the recurrence and non-recurrence groups using ANOVA and OLS regression, we determined that the T-category, N-category, and preoperative CEA levels had a significant relationship with CRC recurrence. These findings confirmed the well-known risk factors for recurrence in this study, and the use of similar adjuvant chemotherapy was unlikely to reduce recurrence. Particularly, the preoperative CEA level supports previous studies that found a higher recurrence or death risk in patients with stage III CRC with high CEA levels [54]. These results emphasize the need for precision medicine tailored to each patient, rather than a uniform adjuvant treatment, considering the T-and N-categories, preoperative CEA levels, and genetic variation. In addition to the analysis, future research may benefit from incorporating broader patient-level factors. Prior studies have demonstrated that certain lifestyle behaviors, including sugary drink and liquid fat consumption, are associated with CRC recurrence risk [55]. Although such variables were not available in our dataset, integrating lifestyle information, comorbidities, and treatment adherence into future analyses could enhance explanatory power.

We identified 221 SSMs between the recurrence and non-recurrence groups and analyzed their biological functions through protein-protein interactions using IPA. Significant characteristics emerge in certain networks. As shown in Figs C, D, G, and I in File S3, the other networks provide clarity. For example, networks 1 and 2 (Figs A and B in File S3) revealed mutations affecting the extracellular signal-regulated kinase (ERK) pathway, which is closely associated with CRC. ERK is one of the three major subfamilies of mitogen-activated protein kinases (MAPK) closely associated with CRC [56,57]. Networks 6 and 8 (Figs F and H in File S3) demonstrated a connection with *CTNNB1* (Table 5), which is linked to CRC

development and metastasis [58,59]. Network 5 (Fig E in S3 File) displayed a notable link to *CCND1*, which is associated with cancer progression and metastasis [60,61]. These findings suggest that gene mutations within this network can negatively affect the expression of *CCND1* or its downstream effects.

While IPA provided a high-confidence framework to interpret the functional context of significant mutations, it may not be enough to explain protein-protein interactions. To address this limitation, we conducted an additional analysis using the STRING database [62] to visualize interactions within networks 5 and 8, which showed the strongest associations with recurrence and non-recurrence of CRC. In this analysis, network 5 revealed a new connectivity pattern involving *SS18L1*, which was sparsely connected in the IPA network but showed multiple interactions in STRING. This finding highlights the potential relevance of *SS18L1*, whose mutation was associated with the non-recurrence group in our data. Furthermore, previous studies have suggested a link between *SS18L1* and CRC [63,64] indicating that further investigation of this SSM may provide novel insights into its role in CRC progression. In contrast, many genes in network 8 remained isolated, suggesting that these genes may be poorly characterized and warrant further investigation. The results of the STRING analysis for networks 5 and 8 are provided in Supplementary File S4.

We also analyzed splice-site mutations, which are known to be significant because they can substantially alter protein structure. We identified mutations in *LILRA2* and *SRPRA* with a high probability of causing splice loss, affecting exon and intron sequences, and predicted protein structures (Figs C-H in the S2 File). In *LILRA2*, Gao *et al.* revealed that high expression is correlated with improved survival without recurrence, while Zhang *et al.* found conflicting results that high expression is associated with poorer cancer prognosis [65,66]. Although our study showed that mutations in *LILRA2* are related to CRC recurrence, previous findings conflict with each other and necessitate further investigation [65]. We suggest analyzing *LILRA2* in conjugation with other genes in the same network, as shown in Fig 6C. Various studies have explored the role of SRP in cancer, but there has been minimal focus on its receptor *SRPRA* [67,68]. Since our research shows a relationship between mutations in the *SRPRA* gene and low CRC recurrence, we suggest further investigation into the SRP receptor in addition to SRP itself.

A recurrence prediction model based on SSMs in protein-protein interaction networks was developed. The high variable importance of networks 8 and 5 (Fig 6A) indicates their significant influence on classification performance. Figs E and H in S3 File (Figs 6C and 6D) align with the non-recurrence and recurrence groups, respectively. This model demonstrated an AUC of 0.77 (Fig 6B), indicating good discriminatory power [69,70]. However, validation is challenging because of the limited sample size, necessitating further studies. However, validation using independent patient cohorts remains challenging due to the lack of publicly available datasets containing tumor-only somatic mutation profiles linked with recurrence outcomes. We acknowledge this limitation and emphasize the importance of future studies involving larger and more diverse cohorts to assess the generalizability of our model. Moreover, broader efforts to generate and release such datasets are essential to support external validation and clinical translation of predictive models.

We validated the effectiveness of third-party databases for classifying variants by comparing their predictions with those of normal samples from a comparable cohort. The confusion matrix in Table 6 achieved a high precision of 0.973 and recall of 0.845, indicating strong performance. Notably, 48.76% of the common variants that were filtered out in this study were not classified as common variants by the normal samples. This indicates a limitation of our study, as it suggests that we may have filtered out mutations potentially associated with CRC recurrence. However, because rigorous variant filtering is recommended for tumor-only samples, this result shows that the use of third-party databases is recommended, even if a comparable cohort exists.

We found that 100 out of the 221 SSMs showed a greater than 50% likelihood of being germline mutations based on their VAFs. This underscores a key limitation of tumor-only sequencing, which cannot distinguish somatic from rare germline variants with certainty. This result also indicates that, despite filtering with population databases, germline variants may remain in the dataset, potentially affecting the interpretation of recurrence-associated mutations. In future studies, tumor-normal paired sequencing will be necessary to validate somatic status and clarify clinical relevance.

We investigated the potential functional impact of the identified SSMs using publicly available gene expression datasets through the Xena platform [71] with TCGA Colon and Rectal Cancer dataset. For this analysis, we focused on the *REXO1* S589del mutation, which was the most frequently observed SSM across both recurrence and non-recurrence groups. This mutation was selected because the public gene expression dataset we used was not sufficiently large and did not match with our Korean cohort to allow evaluation of less frequent mutations. In the case of *REXO1*, the data indicated that the sample carrying the S589del mutation exhibited elevated *REXO1* expression compared to non-mutated samples with the log2(norm count+1) value of 10.2, suggesting a potential transcriptional impact. Interestingly, previous study has reported a positive correlation between *REXO1* and *ARID3A* expression, the latter of which upregulates *AUCKR*, a factor implicated in CRC progression [72]. Given that the *REXO1* S589del mutation was strongly associated with recurrence (OR = 3.51), this finding suggests a mechanistic link between the mutation and recurrence risk, consistent with prior biological findings. Although this result does not substitute for direct experimental validation, it can support for the biological relevance of the *REXO1* SSM and emphasize the need for further investigation. Unfortunately, we could not find gene expression changes for most other SSMs we found in this research. To overcome this limitation, the availability of sufficiently large public datasets becomes essential for meaningful interpretation.

Consequently, we identified 221 mutations with statistically significant differences between the recurrence and non-recurrence groups in Korean CRC tumor-only samples. Using IPA, we explored gene-level relationships across nine protein-protein interaction networks and implemented a CRC recurrence prediction model. The necessity of using various databases to filter common variants in tumor-only sample analyses was demonstrated. Our study also highlights the limitations of the current Korean databases, suggesting the need for enhancements or new developments to improve research accuracy and relevance.

Our study focused on tumor-only whole-exome sequencing data to reflect the type of genomic information often available in real-world research and clinical settings. However, whole-exome sequencing has inherent limitations. Whole-exome sequencing does not reliably detect large structural variations, copy number alterations, or epigenetic modifications, all of which may contribute significantly to CRC recurrence. Although recent methods attempt to infer copy number alterations from whole-exome sequencing data [73], the precision of current approaches remains limited, especially in the context of tumor-only sequencing where matched normal references are unavailable. Therefore, future developments in analytical tools that expand the resolution of tumor-only whole-exome sequencing to include broader genomic alterations will be critical for achieving more comprehensive recurrence risk stratification.

In addition, due to the absence of matched normal samples, our analysis could not account for germline variants that may underlie hereditary CRC syndromes such as Lynch syndrome [74] or familial adenomatous polyposis [75]. These germline variants are important for improving risk stratification and guiding patient management. Although this study could not assess germline variants, future investigations incorporating tumor-normal paired sequencing may help clarify the complementary roles of somatic and germline mutations in CRC development and recurrence. Furthermore, if robust methodologies are developed to reliably infer germline variants from whole-exome sequencing data, such approaches would be highly valuable, particularly in clinical or research settings where matched normal tissue is unavailable.

## Supporting information

**S1 Table. Summary data of samples sequenced.**
(XLSX)

**S2 Table. Numbers of patients with mean and standard deviation for each clinical variable.**
(XLSX)

**S3 Table. List of statistically significant mutations (SSMs).**
(XLSX)

**S4 Table. Information of SSMs in splicing sites.** Mutation information, odds ratios, and SpliceAI model delta scores of splice site mutations in *LILRA2, FYB1, HACL1, RPRD1A, NCAPD3, SRPRA, SERPINB6* genes. In the exon sequence (protein) column, lowercase letters denote nucleotides and uppercase letters represent amino acids.
(XLSX)

**S5 Table. Result of SSMs in splicing sites using SpliceAPP.** Expected intron effected by the splicing site SNVs with the calculated effect using Splice. APP.
(XLSX)

**S6 Table. Model performance comparison.**
(XLSX)

**S7 Table. Estimated germline mutation probability for each SSM across sample groups.**
(XLSX)

**S1 File. Mutation profile of recurrence group and non-recurrence group after sample filtering.**
(DOCX)

**S2 File. SSMs in splicing sites cause splice loss.**
(DOCX)

**S3 File. Protein-protein interaction analyzed by IPA.** Orange circles indicate a higher frequency of mutations in the recurrence group, blue circles denote a higher frequency of mutations in the non-recurrence group, and purple circles represent cases in which the mutations were significant in both groups.
(DOCX)

**S4 File. Protein-protein interaction analyzed by STRING.** Orange circles indicate a higher frequency of mutations in the recurrence group, and blue circles denote a higher frequency of mutations in the non-recurrence group.
(DOCX)

**S1 Figure. Bar plot of amino acid substitutions in nonsynonymous SNVs.** A total of 77,178 non-synonymous SNVs were identified. Amino acid substitutions that appeared in over 3,000 variants were colorized red (R > Q, R > H, and A > T), over 2,500 variants were colorized orange (R > C and A > V), and over 2,000 variants were colorized yellow (R > W and P > L). The others are colored in grey.
(TIF)

We would like to thank the Computer System Team at KOBIC for providing and managing the infrastructure for the research analysis. We also thank Jinseon Yoo at KOBIC for his thoughtful input and helpful suggestions on methodological aspects.

## Author contributions

**Conceptualization:** Hajin Jeon, Jong Lyul Lee, Jin Ok Yang.

**Data curation:** Hajin Jeon, Jong Lyul Lee, Hyeran Shim, Soobok Joe, Chan Wook Kim, Seok-Byung Lim, In Ja Park, Yong Sik Yoon, Hoang Bao Khanh Chu, Young-Joon Kim, Chang Sik Yu.

**Formal analysis:** Hajin Jeon.

**Funding acquisition:** Jong Lyul Lee, Young-Joon Kim, Chang Sik Yu, Jin Ok Yang.

**Investigation:** Hajin Jeon, Jong Lyul Lee, Jin Ok Yang.

**Methodology:** Hajin Jeon, Jin Ok Yang.

**Project administration:** Hajin Jeon, Young-Joon Kim, Chang Sik Yu, Jin Ok Yang.

**Resources:** Jong Lyul Lee, Hyeran Shim, Iksu Byeon, Hoang Bao Khanh Chu, Young-Joon Kim, Chang Sik Yu, Jin Ok Yang.

**Software:** Hajin Jeon.

**Supervision:** Hajin Jeon, Jong Lyul Lee, Young-Joon Kim, Jin Ok Yang.

**Validation:** Hajin Jeon, Jong Lyul Lee, Jin Ok Yang.

**Visualization:** Hajin Jeon.

**Writing – original draft:** Hajin Jeon.

**Writing – review & editing:** Hajin Jeon, Jong Lyul Lee, Hyeran Shim, Soobok Joe, Young-Joon Kim, Chang Sik Yu, Jin Ok Yang.

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
