## [Decision Letter · Decision Letter 0]

11 Feb 2025

PONE-D-24-54119Genetic Variations and Recurrence in Stage III Korean Colorectal Cancer: Insights from Tumor-Only Mutation AnalysisPLOS ONE

Dear Dr.  Yang,

Thank you for submitting your manuscript to PLOS ONE. After careful consideration, we feel that it has merit but does not fully meet PLOS ONE’s publication criteria as it currently stands. Therefore, we invite you to submit a revised version of the manuscript that addresses the points raised during the review process.

We look forward to receiving your revised manuscript.

Kind regards,

Nejat Mahdieh

Academic Editor

PLOS ONE

Journal Requirements:

Reviewers' comments:

Reviewer's Responses to Questions

**Comments to the Author**

1. Is the manuscript technically sound, and do the data support the conclusions?

Reviewer #1: Partly

Reviewer #2: Yes

2. Has the statistical analysis been performed appropriately and rigorously? 

Reviewer #1: Yes

Reviewer #2: Yes

3. Have the authors made all data underlying the findings in their manuscript fully available?

Reviewer #1: Yes

Reviewer #2: Yes

4. Is the manuscript presented in an intelligible fashion and written in standard English?

Reviewer #1: Yes

Reviewer #2: Yes

5. Review Comments to the Author

Reviewer #1: Dear Editor,

I appreciate the opportunity to review the manuscript PONE-D-24-54119, titled "Genetic Variations and Recurrence in Stage III Korean Colorectal Cancer: Insights from Tumor-Only Mutation Analysis." This study addresses a critical clinical question regarding the genetic factors contributing to colorectal cancer (CRC) recurrence in stage III patients. The authors have employed whole-exome sequencing (WES) and machine learning models to identify potential genetic markers of recurrence, making a valuable contribution to precision oncology. The study is well-designed and has several major strengths that enhance its scientific and clinical impact. However, I have the following major concerns:

1.The study utilizes tumor-only sequencing, which does not differentiate between somatic and germline mutations. This may result in the misclassification of inherited variants as tumor-specific, potentially affecting clinical interpretations.

2.The study focuses exclusively on Korean patients with stage III colorectal cancer. Given that genetic and environmental factors influencing recurrence may differ across ethnic groups and CRC stages, the applicability of these findings to broader populations is limited. Please discuss this limitation.

3.While the study identifies statistically significant mutations, it does not confirm their functional impact through experimental validation (e.g., gene expression studies, protein function assays). Without biological validation, the clinical relevance of these mutations remains uncertain.

4.The CRC recurrence prediction model, developed using machine learning (PyCaret), achieved an AUC of 0.77. While this suggests moderate predictive performance, further validation with external patient cohorts is necessary before clinical application.

5.The study considers clinical variables such as T-category, N-category, and preoperative carcinoembryonic antigen (CEA) levels but does not account for lifestyle factors, treatment adherence, or comorbidities, which may also influence CRC recurrence. Including such factors would provide a more comprehensive analysis.

6.The study excludes tumors with >24 mutations per megabase to remove hypermutated samples. However, hypermutation status plays a crucial role in immune response and tumor evolution. Its exclusion may limit insights into the mechanisms of CRC recurrence.

I appreciate the authors' efforts and believe addressing these concerns will further strengthen the study.

Reviewer #2: The study provides valuable insights into the genetics of CRC recurrence; however, it would benefit from functional validation, external cohort validation, and broader population studies. I recommend considering minor revisions to enhance methodological robustness and clinical applicability. Below are my comments:

•While the study filters common variants using gnomAD, ExAC, dbSNP, 1000 Genomes Project, KOVA2, and KRGDB, how do the authors address database inaccuracies or population-specific variants that may lead to false positives (wrongly retained somatic variants) or false negatives (important tumor mutations mistakenly removed)?

•WES cannot reliably detect large structural variations, copy number alterations (CNVs), or epigenetic modifications, which may be critical in CRC recurrence. Could the authors provide explanations or alternative approaches to account for these limitations?

•The study utilizes Ingenuity Pathway Analysis (IPA) for protein-protein interaction (PPI) network analysis, which relies on pre-existing knowledge. However, novel mutations and unknown interactions may be overlooked without experimental validation (e.g., CRISPR functional studies, RNA-seq for expression analysis). Could the authors discuss the potential impact of this limitation?

•The study identifies seven splicing site mutations and predicts their effects using SpliceAI. However, without RNA sequencing (RNA-seq) validation, it remains unclear whether these mutations actually result in abnormal splicing events in CRC patients. How do the authors address this concern?

•An important limitation of this study is its exclusive focus on somatic mutations in tumor samples, without assessing germline risk variants that may contribute to hereditary CRC syndromes (e.g., Lynch syndrome, APC mutations in FAP). Acknowledging germline contributions could enhance risk stratification and patient management strategies. Could the authors discuss how future studies might address this gap?

6. PLOS authors have the option to publish the peer review history of their article (what does this mean? ). If published, this will include your full peer review and any attached files.

**Do you want your identity to be public for this peer review?** For information about this choice, including consent withdrawal, please see our Privacy Policy .

Reviewer #1: **Yes: ** Amirhassan Rabbani

Reviewer #2: No

---

## [Author Response · Author response to Decision Letter 0]

1 Apr 2025

We would like to express our sincere gratitude to the editor and reviewers for their thorough and constructive feedback on our manuscript. We have carefully considered all comments and have revised the manuscript accordingly. Below, we provide a detailed point-by-point response to each comment, outlining the corresponding revisions and justifications.

Sincerely,

The Authors of PONE-D-24-54119

------

To the Editor,

(1) We have prepared a detailed, point-by-point rebuttal letter in response to all comments raised by the academic editor and reviewers. This document has been uploaded as a separate file labeled "Response to Reviewers".

(2) A marked-up version of the revised manuscript, showing all changes made to the original version, has been uploaded as "Revised Manuscript with Track Changes." All modifications are indicated in red font, and the Microsoft Word track changes feature has also been used.

(3) An unmarked version of the revised manuscript, without tracked changes, has been uploaded as "Manuscript".

We have thoroughly rechecked our manuscript to ensure compliance with PLOS ONE’s style guidelines. In particular, we have corrected file names to match the journal’s conventions (e.g., “Response to Reviewers,” “Revised Manuscript with Track Changes,” “Manuscript”). We apologize for any previous inconsistencies and have now resolved these issues.

We acknowledge that our initial repository (KoNA) does not meet PLOS ONE’s standards. To address this, we have revised our Data Availability Statement to clarify that our dataset is currently under restricted access due to national research program requirements. However, it will be publicly released through the Korean BioData Station (K-BDS; https://kbds.re.kr/) under the accession IDs KAP220472, KAP220473, and KAP230611. Researchers may request early access via the K-BDS platform according to applicable guidelines.

Accordingly, the original Data Availability Statement:

“The raw data were deposited in the Korean Nucleotide Archive (KoNA) under accession IDs KAP220472, KAP220473 and KAP230611.”

has been revised to the following:

“Data cannot be shared publicly because of legal and ethical restrictions related to the national research program under which the data were generated. The data are scheduled for public release through the Korean BioData Station (https://kbds.re.kr/) under accession numbers KAP220472, KAP220473, and KAP230611. Until public release, access may be requested via the K-BDS platform. We respectfully request an exemption from the open data policy until the data are released according to this schedule”

We hope this revised statement meets the journal’s requirements and provides a clear path for data access and future availability.

Thank you for the guidance. We have uploaded all figures included in the manuscript to the PACE system and used the processed output files in the submission, in accordance with PLOS figure requirements.

-----

To the Reviewer 1,

1.The study utilizes tumor-only sequencing, which does not differentiate between somatic and germline mutations. This may result in the misclassification of inherited variants as tumor-specific, potentially affecting clinical interpretations.

We appreciate this important observation. Although we applied strict filtering using multiple population databases (gnomAD, ExAC, 1000 Genomes, etc.) to reduce germline contamination, we acknowledge that some germline variants may remain. In the revised manuscript, we have: Conducted an additional variant allele frequency (VAF) analysis to estimate the likelihood of each somatic single nucleotide mutation (SSM) being germline in origin. Determined that approximately 100 out of 221 SSMs had a >50% probability of being germline. Added explicit cautions in the Introduction, Methods, Results, and Discussion to highlight this limitation. Included a new Supplementary Table S7 summarizing these findings.

2.The study focuses exclusively on Korean patients with stage III colorectal cancer. Given that genetic and environmental factors influencing recurrence may differ across ethnic groups and CRC stages, the applicability of these findings to broader populations is limited. Please discuss this limitation.

We agree that restricting the cohort to Korean stage III CRC patients may limit generalizability to other ethnicities or CRC stages. However, Korea has a relatively homogeneous population, and a significant proportion of CRC-related deaths occur in stage III patients here. We have clarified in the Introduction that our aim was to address a critical issue in this specific national context, but we also acknowledge the importance of validating these findings in more diverse populations.

3.While the study identifies statistically significant mutations, it does not confirm their functional impact through experimental validation (e.g., gene expression studies, protein function assays). Without biological validation, the clinical relevance of these mutations remains uncertain.

We concur that experimental validation (e.g., gene expression or protein function assays) would strengthen our findings. While new experiments were not feasible within this study’s scope, we performed a limited in silico validation by examining available public datasets for key genes (e.g., REXO1). We found indications (e.g., correlation with ARID3A) suggesting a functional role, but emphasize that further biological confirmation is necessary. We have added a paragraph in the Discussion regarding this limitation.

4.The CRC recurrence prediction model, developed using machine learning (PyCaret), achieved an AUC of 0.77. While this suggests moderate predictive performance, further validation with external patient cohorts is necessary before clinical application.

We agree. Due to the lack of publicly available tumor-only recurrence datasets, we could not perform external validation. We have acknowledged this shortcoming in the Discussion and highlighted the need for independent cohorts. As more datasets become available, we plan to refine and validate our model.

5.The study considers clinical variables such as T-category, N-category, and preoperative carcinoembryonic antigen (CEA) levels but does not account for lifestyle factors, treatment adherence, or comorbidities, which may also influence CRC recurrence. Including such factors would provide a more comprehensive analysis.

We appreciate this suggestion. Our dataset focused primarily on molecular and core clinical variables from medical records, and data on lifestyle or comorbidities were not available. We have acknowledged this limitation in the Discussion, citing relevant research that underscores the importance of such factors. We recommend that future studies incorporate more comprehensive clinical variables.

6.The study excludes tumors with >24 mutations per megabase to remove hypermutated samples. However, hypermutation status plays a crucial role in immune response and tumor evolution. Its exclusion may limit insights into the mechanisms of CRC recurrence.

I appreciate the authors' efforts and believe addressing these concerns will further strengthen the study.

We initially excluded 13 hypermutated cases (5 recurrence, 8 non-recurrence) for consistency in our main analysis. To address this concern, we performed a supplementary analysis on these 13 samples (see Supplementary File S1, Tables C–D and Figures E–H). These results are also summarized in the revised Methods, Results, and Discussion. We acknowledge the potential importance of hypermutated tumors and recommend future studies with larger hypermutated cohorts.

-----

To the Reviewer 2,

1. While the study filters common variants using gnomAD, ExAC, dbSNP, 1000 Genomes Project, KOVA2, and KRGDB, how do the authors address database inaccuracies or population-specific variants that may lead to false positives (wrongly retained somatic variants) or false negatives (important tumor mutations mistakenly removed)?

To minimize false positives, we used the union of multiple databases (gnomAD, ExAC, dbSNP, 1000 Genomes, KOVA2, KRGDB). We acknowledge that some true mutations might be filtered out (false negatives), but this approach increases confidence that retained mutations are truly somatic. We have clarified this rationale in the revised manuscript.

2. WES cannot reliably detect large structural variations, copy number alterations (CNVs), or epigenetic modifications, which may be critical in CRC recurrence. Could the authors provide explanations or alternative approaches to account for these limitations?

We agree WES has limited capacity for detecting structural variations, CNVs, or epigenetic modifications. While tools like ECOLE exist to estimate CNVs from WES, their accuracy remains insufficient for clinical use. We highlight these limitations in the revised Discussion and emphasize the need for expanded genomic profiling methods.

3. The study utilizes Ingenuity Pathway Analysis (IPA) for protein-protein interaction (PPI) network analysis, which relies on pre-existing knowledge. However, novel mutations and unknown interactions may be overlooked without experimental validation (e.g., CRISPR functional studies, RNA-seq for expression analysis). Could the authors discuss the potential impact of this limitation?

IPA draws on a curated database, potentially missing novel interactions. To partially address this, we cross-referenced results with the STRING database. Notably, SS18L1 (associated with non-recurrence) formed more connections in STRING than in IPA, suggesting further experimental study is warranted. We discuss these points in the revised Discussion and include STRING-based network visualizations in Supplementary File S4.

4. The study identifies seven splicing site mutations and predicts their effects using SpliceAI. However, without RNA sequencing (RNA-seq) validation, it remains unclear whether these mutations actually result in abnormal splicing events in CRC patients. How do the authors address this concern?

We acknowledge that RNA-seq validation is the gold standard to confirm splicing defects. Due to data and IRB constraints, we could not conduct such experiments. As an alternative, we used SpliceAPP alongside SpliceAI; both tools independently supported potential splicing effects for mutations in LILRA2 and SRPRA, though they differed on SERPINB6. These findings are included in the revised Results and Supplementary Table S5.

5. An important limitation of this study is its exclusive focus on somatic mutations in tumor samples, without assessing germline risk variants that may contribute to hereditary CRC syndromes (e.g., Lynch syndrome, APC mutations in FAP). Acknowledging germline contributions could enhance risk stratification and patient management strategies. Could the authors discuss how future studies might address this gap?

Germline risk variants (e.g., Lynch syndrome, FAP) are crucial but could not be systematically assessed without matched normal samples. We now explicitly state this limitation in the Discussion, noting that future tumor-normal paired studies are needed for comprehensive hereditary risk analysis.

---

## [Decision Letter · Decision Letter 1]

7 Apr 2025

Genetic Variations and Recurrence in Stage III Korean Colorectal Cancer: Insights from Tumor-Only Mutation Analysis

PONE-D-24-54119R1

Dear Dr. Yang,

We’re pleased to inform you that your manuscript has been judged scientifically suitable for publication and will be formally accepted for publication once it meets all outstanding technical requirements.

Kind regards,

Nejat Mahdieh

Academic Editor

PLOS ONE

Additional Editor Comments (optional):

Reviewers' comments:

Reviewer's Responses to Questions

**Comments to the Author**

1. If the authors have adequately addressed your comments raised in a previous round of review and you feel that this manuscript is now acceptable for publication, you may indicate that here to bypass the “Comments to the Author” section, enter your conflict of interest statement in the “Confidential to Editor” section, and submit your "Accept" recommendation.

Reviewer #3: All comments have been addressed

2. Is the manuscript technically sound, and do the data support the conclusions?

Reviewer #3: Yes

3. Has the statistical analysis been performed appropriately and rigorously? 

Reviewer #3: (No Response)

4. Have the authors made all data underlying the findings in their manuscript fully available?

Reviewer #3: Yes

5. Is the manuscript presented in an intelligible fashion and written in standard English?

Reviewer #3: Yes

6. Review Comments to the Author

Reviewer #3: (No Response)

7. PLOS authors have the option to publish the peer review history of their article (what does this mean? ). If published, this will include your full peer review and any attached files.

**Do you want your identity to be public for this peer review?** For information about this choice, including consent withdrawal, please see our Privacy Policy .

Reviewer #3: **Yes: ** Amirhassan Rabbani

---

## [Editor Report · Acceptance letter]

PONE-D-24-54119R1

PLOS ONE

Dear Dr. Yang,

I'm pleased to inform you that your manuscript has been deemed suitable for publication in PLOS ONE. Congratulations! Your manuscript is now being handed over to our production team.

Kind regards,

on behalf of

Dr. Nejat Mahdieh

Academic Editor

PLOS ONE